# Single-electron induced surface plasmons on a topological nanoparticle

G. Siroki[1], D.K.K. Lee[1], P.D. Haynes[1,2] & V. Giannini[1]

It is rarely the case that a single electron affects the behaviour of several hundred thousands of atoms. Here we demonstrate a phenomenon where this happens. The key role is played by topological insulators—materials that have surface states protected by time-reversal symmetry. Such states are delocalized over the surface and are immune to its imperfections in contrast to ordinary insulators. For topological insulators, the effects of these surface states will be more strongly pronounced in the case of nanoparticles. Here we show that under the influence of light a single electron in a topologically protected surface state creates a surface charge density similar to a plasmon in a metallic nanoparticle. Such an electron can act as a screening layer, which suppresses absorption inside the particle. In addition, it can couple phonons and light, giving rise to a previously unreported topological particle polariton mode. These effects may be useful in the areas of plasmonics, cavity electrodynamics and quantum information.

[1] Department of Physics, Imperial College London, Prince Consort Road London, London SW7 2AZ, UK. [2] Department of Materials, Imperial College London, London SW7 2AZ, UK. Correspondence and requests for materials should be addressed to V.G. (email: v.giannini@imperial.ac.uk).

Topologically protected surface states have many unusual properties and today their analogues are also investigated in photonics[1,2], acoustics[3,4] and optical lattices[5]. In solid-state physics, the associated materials that have attracted a lot of attention recently are known as topological insulators (TIs)[6]. TIs possess gapless and delocalized surface states[7,8] protected by time-reversal symmetry. This symmetry implies that the surface states of TIs cannot be localized or removed by defects and non-magnetic impurities as opposed to the case of ordinary insulators[7]. Furthermore, the surface states have shown very little temperature dependence[9,10] and are also observed at room temperature[9–11]. They were previously studied in the case of bulk TIs terminated at an arbitrary crystal face[12], thin films[13], cylinders[14–16] and spheres[17–19]. Here we study how the surface states affect optical properties of TI nanoparticles (TINPs) with sizes below $\sim 100$ nm. Apart from ref. 20, previous studies of the optical properties of TIs were restricted to flat surfaces. In particular, the predicted topological magnetoelectric effect[8] led to experimentally observed strong Kerr rotation of light reflected by thin films of TIs[21], which were also predicted to absorb light strongly[22]. Furthermore, surface Dirac plasmons were found to modify extinction in TI microribbons[23,24].

In this work we theoretically investigate how surface states affect the optical properties of nanoscale-size TIs. We focus on a single spherical TINP made of $Bi_2Se_3$, a prototype TI material due to its relatively large gap of 0.3 eV. In analogy with the previous studies[23,24] we find that the extinction is modified due to the interaction of a bulk phonon with surface electrons (implying Dirac fermions in the topologically protected surface states). However, in contrast to the bulk samples[23,24], for particle radii below $\sim 100$ nm individual electrons start playing an important role and cannot be described collectively by a Drude term but require a quantum-mechanical treatment. In TINPs the spectrum of surface states is not continuous but discrete, similar to ordinary quantum dots with the important difference that electrons are delocalized over the surface. This leads to qualitatively different behaviour compared with the bulk samples. Surprisingly, even a single electron in a topologically protected surface state can lead to surface charge density that affects the whole nanoparticle. Such an electron screens the body of the nanoparticle from external field at certain frequencies. Furthermore, it mediates the interaction between a bulk phonon and light to produce a previously unreported mode. The mode was not revealed in earlier studies, which focused on bulk samples and thin films of TIs. This surface topological particle (SToP) mode couples strongly to the localized surface plasmon polariton (LSPP).

## Results

**Ordinary insulator nanoparticle.** To start with, it is instructive to calculate absorption of a bare nanoparticle. We consider a single spherical $Bi_2Se_3$ nanoparticle. To keep the solution analytic we take the **c** axis of the material, which has a layered structure, to be parallel to the photon wavevector **k**—**c**∥**k**. The particle is in vacuum and is illuminated by light of frequency $w/2\pi$ (see Fig. 1a). We work in the quasi-static limit, because the particle radius is much smaller than the wavelength of light, $\lambda \gg R$. The boundary conditions on the electric field determine the absorption cross-section (Supplementary Note 1) given by

$$\sigma_{abs} = 4\pi R^3 \frac{2\pi}{\lambda} \mathrm{Im} \left[ \frac{\epsilon_{in} + \delta_R - 1}{\epsilon_{in} + \delta_R + 2} \right] \quad (1)$$

where $\epsilon_{in}$ is the bulk dielectric function (Fig. 1b) and the term $\delta_R$ arises due to transitions between the delocalized topologically protected surface states perturbed by the incident light. If this contribution is removed, $\delta_R = 0$, then equation (1) reduces to the

textbook result of a dielectric sphere in a uniform field. Its optical response follows from the spherical shape and the bulk dielectric function of $Bi_2Se_3$ (see Fig. 1b) modelled by

$$\epsilon_{in}(w) = \sum_{j=\alpha,\beta,f} \frac{w_{pj}^2}{w_{0j}^2 - w^2 - i\gamma_j w} \quad (2)$$

which contains contributions from $\alpha$ and $\beta$ transverse phonons, as well as free charge carriers (labelled $f$) arising from the bulk defects. The parameters for the three terms in equation (2) are taken from a fit to experimental data on bulk $Bi_2Se_3$ (ref. 25) and presented in Table 1.

To summarize, in our model we treat light and the collective response from the body of the nanoparticle classically, while the surface states are found using a low-energy Hamiltonian valid close to the Dirac point[17,26]. Without the surface states ($\delta_R = 0$) the particle acts as an ordinary insulator. In particular, its absorption cross-section contains two peaks at 1.05 and 3.72 THz as shown in Fig. 1d (blue dashed line). The one at lower energy is caused by the LSPP due to free bulk charge carriers and the one at higher energy is due to the $\beta$ phonon. In contrast, adding surface states to the model results in a non-zero $\delta_R$ and modified cross-section. The result shown in Fig. 1d (red solid line) is the main finding of this work. It contains an additional peak in absorption and a point of zero absorption (see arrow in Fig. 1d), which are further discussed in the next section.

**Topological insulator nanoparticles.** Let us now see how the absorption cross-section of the nanoparticle is modified by the occupied topologically protected surface states. In the presence of time reversal symmetry, the existence of surface states is guaranteed by the non-trivial topology of bulk band structure. In the case of a flat $Bi_2Se_3$ surface, this results in a single Dirac cone at the $\Gamma$—point in the reciprocal space. The Dirac cone covers the area $A_{\mathbf{k}}$, which is a small fraction of the two-dimensional first Brillouin zone. In real space their wavefunctions cannot be localized over areas smaller that $\approx (2\pi)^2 / A_{\mathbf{k}}$, rendering them insensitive to atomic size details of the surface as noted in ref. 19 (and references therein). This is even more so for the nanoparticle surface where the few discrete states available cannot produce fine features in real space. As a result, the surface states are influenced by two key factors: surface termination and curvature. It has been demonstrated that the Dirac cone is only slightly modified for different terminations[12]. It is also known that the surface curvature can be taken into account explicitly through the Berry phase[14–17]. This allows us to employ a simplified analytical model of a spherical TINP derived by Imura *et al.*[17].

Here we apply time-dependent perturbation theory to the model of ref. 17, which, for small radii, yields a discretized Dirac cone on spherical surface. For $Bi_2Se_3$ the resulting surface states have energies $\pm A/R$, $\pm 2A/R$ and so on (see Fig. 1c), where $A = 3.0$ eV·Å is a constant obtained from density-functional theory calculations[26] that arises due to spin–orbit coupling and enters the surface Dirac equation. The analytical model can be tested using tight-binding calculations and past studies for cylindrical[15,16] and cubic[17] TIs showed good agreement between the two approaches. To further test the model employed, we performed tight-binding calculations for a spherical particle using parameters from ref. 26—the results are shown in Fig. 2 below. The agreement improves for large radii as the surface of the spherical cluster modelled with tight-binding becomes more smooth. Close agreement for the cubic particle of the same volume is due to topological nature of the surface states. Both tight-binding and analytical models predict the same degeneracy of the surface states (Supplementary Fig. 1). They break down for small volumes when there is insufficient bulk material to support

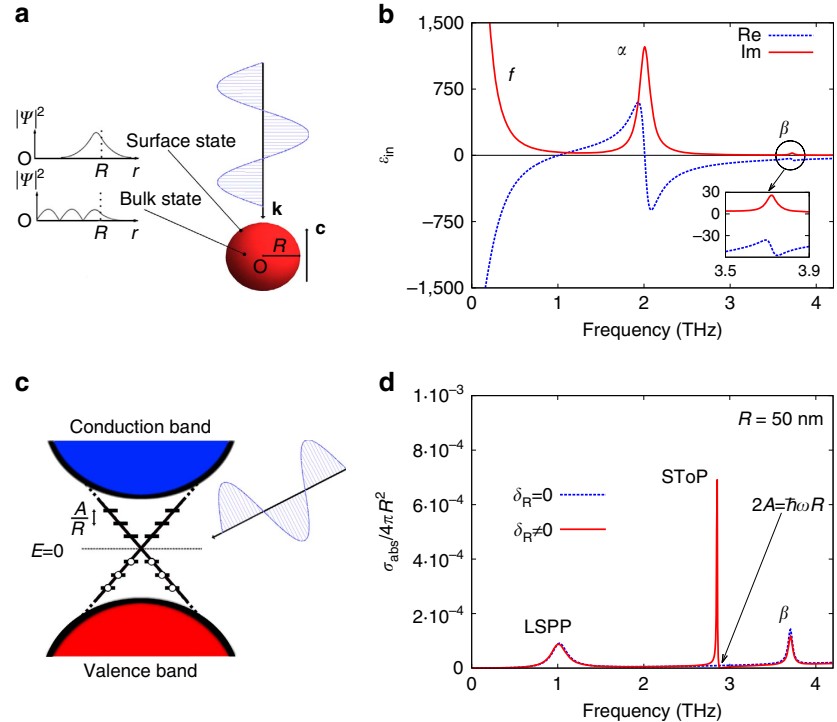

**Figure 1 | TINP interacting with light** (**a**) The system studied: a spherical $Bi_2Se_3$ nanoparticle of radius $R$ located at the origin is subject to an EM wave propagating along its **c** axis—**k**||**c**. The collective response of the bulk states is described using the dielectric function $\epsilon_{in}$, whereas the surface states are treated quantum mechanically. (**b**) The dielectric function $\epsilon_{in}$ of $Bi_2Se_3$ comprises two phonons at 2.0 ($\alpha$) and 3.72 THz ($\beta$), as well as a contribution from bulk free charge carriers (*f*) at low frequencies[25]. (**c**) In the nanoparticle, the surface Dirac cone consists of discrete levels symmetrically placed with respect to the Dirac point ($E = 0$)[17]. They have constant energy spacing $A/R$ where $A = 3.0$ eV $\cdot$ Å. An electron in a surface state couples to other states under the influence of light producing a time-dependent surface charge density. (**d**) The absorption cross-section of a bare nanoparticle ($\delta_R = 0$, blue dashed line) is dictated by the dielectric function of the material. It is modified when electrons occupy delocalized topological surface states ($\delta_R \neq 0$, red solid line). The resultant spectrum contains an additional mode (SToP) and a zero in absorption (marked with an arrow).

**Table 1 | Parameters for the bulk dielectric function of $Bi_2Se_3$.**

|  | $w_{pj}$ | $w_{0j}$ | $\gamma_j$ |
|---|---|---|---|
| $\alpha$ | 19.2 | 2.0 | 0.15 |
| $\beta$ | 2.3 | 3.72 | 0.06 |
| *f* | 11.5 | 0 | 0.24 |

The values given are in THz and were measured at 6 K (ref. 25). The $\alpha$ and $\beta$ phonons are represented by Lorentz terms. The Drude term is due to free bulk charge carriers (*f*) and not surface electrons, which we treat quantum mechanically.

the protected surface states. One way to see this is through the decay constant of surface states into the bulk, which in the present model is $0.5$ nm$^{-1}$ (Supplementary Note 2). Thus, for thicknesses of below a few nanometres the states on opposite surfaces will hybridize strongly. Experimentally, this is observed for $Bi_2Se_3$ thin films of thickness below 5 nm[13]. This is also the radius of the spherical particle at which the energies of the surface states approach the bulk band gap—see shaded region in Fig. 2. This agrees with the behaviour found in TI nanocylinders[15], suggesting that 10 nm radius is a reasonable lower limit at which the model is still valid.

To model the effect of the surface states we suppose that they are occupied up to the Dirac point ($E = 0$) as shown in Fig. 1c. This captures the essential physics and yet keeps the model simple. We focus on the excitation from the highest occupied states at energy $-A/R$, to the lowest vacant states at $A/R$. Dipole transitions from lower occupied states are possible but require energy of at least $4A/R$, which will only be significant for $R$ higher than $\sim 100$ nm; hence, we neglect them. Away from

resonance, this coupling of the surface states yields a time-dependent surface charge density—the surface topological plasmon. It is represented by the term $\delta_R$ in equation (1), which is given by (Supplementary Note 1)

$$\delta_R = \frac{e^2}{6\pi\epsilon_0}\left(\frac{1}{2A - \hbar wR} + \frac{1}{2A + \hbar wR}\right) \quad (3)$$

This term becomes negligible for bulk samples as $R \to \infty$ but is significant for nanoparticles. For small $R$, the spacing becomes comparable to the energy of the optical phonons and the surface state can mediate the interaction between the $\alpha$ phonon and light. The resultant absorption cross-section is modified as shown in Fig. 1d and contains two important features. One is a zero in absorption at a frequency when the denominator of the first term in equation (3) becomes zero. At this point the field inside the particle becomes vanishingly small, to keep the surface charge density finite. Interpreting this result in classical terms, the body of the nanoparticle is being screened by the surface, which becomes perfectly conducting. As a result, the absorption

cross-section becomes zero (the scattering cross-section remains finite but small). The second notable feature in Fig. 1d is the SToP mode, which arises due to the surface topological plasmon coupling to the $\alpha$-phonon and depends on the TINP radius through equation (3).

The results described above are expanded in Fig. 3a where the absorption cross-section is plotted for different particle radii ($R^{-1}$ serves as a wavevector, because the surface states' energy spacing is $A/R$) and can be compared with that of a bare nanoparticle in Fig. 3c. In addition to the two horizontal modes (LSPP and $\beta$-phonon) present in the bare particle, the screening effect and SToP mode now appear. The SToP mode arises due to the interaction of the electrons in the surface states with the bulk $\alpha$ phonon. This is hinted by the fact that the mode is much thinner at the frequency of the $\alpha$ phonon and is further demonstrated in Fig. 3b, where we have removed the $\beta$-phonon

and free carrier terms from the dielectric function (equation (2)) and the SToP mode is practically unchanged. The only difference is due to the fact that when the LSPP and $\beta$-phonon are present, they interact with the SToP mode causing a splitting.

The Rabi splitting is particularly strong if we raise the Fermi energy such that the two surface states immediately above the Dirac point (at $A/R$) become occupied. The resulting cross-section is shown in Fig. 4a. The term $\delta_R$ is now different, although again depends only on the two uppermost electrons. As these electrons couple to the states at energy $2A/R$, the line of suppressed absorption is now at lower frequency and so is the SToP mode. The SToP modes couples to the LSPP as shown in Fig. 4b, with the ratio of the Rabi frequency to that of the peak being $\Omega_R/w > 0.1$, which implies that the coupling is strong. Finally, we note that usually the field inside a material is small whenever $\epsilon_{in}$ is large. This is not the case for the TINP, because the absorption caused by the SToP mode occurs when the real part of denominator in equation (1) is small. Thus, inside the TINP both the field and $\epsilon_{in}$ are considerable, leading to large electromagnetic energy density.

It should be stressed that we have studied the simplest example of the discussed phenomenon. The spherical geometry was chosen, because this case can be solved analytically and provides physical insight. However, the effect will hold for other geometries analogously to the plasmonic resonance, which occurs in metal nanoparticles of spherical and many other shapes[27,28]. This can be explored further by considering more surface states, particles of different shapes and particles that interact with each other. Furthermore, the properties of the TINPs can be modified by placing them in a dielectric matrix and forming regular arrays to create a metamaterial. The surface wavefunctions will be modified by the surrounding medium (or defects and impurities on the surface) compared with the case of the TINP in vacuum, but will stay delocalized as long as time-reversal symmetry is preserved.

**Finite temperature effects.** So far we implicitly assumed that our system is at absolute zero. For practical applications, it is important to consider the effects of scattering caused by phonons at finite temperature. These were studied extensively in $Bi_2Se_3$ in

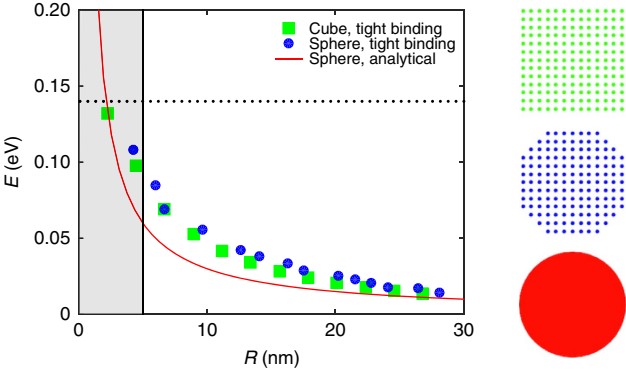

**Figure 2 | Surface states obtained with different methods.** Energy of the first surface state above the Dirac point calculated with a tight-binding model for cubic (green squares) and spherical (blue circles) particles compared with analytical results (red curve). Models are illustrated on the side. Good agreement between the energies of spherical and cubic (edge size $R\sqrt[3]{4\pi/3}$) particles is due to topological nature of the surface states. The horizontal dotted line denotes the bulk band gap and the shaded region is where the model breaks down. Subsequent calculations use $R > 10$ nm where model shows good validity.

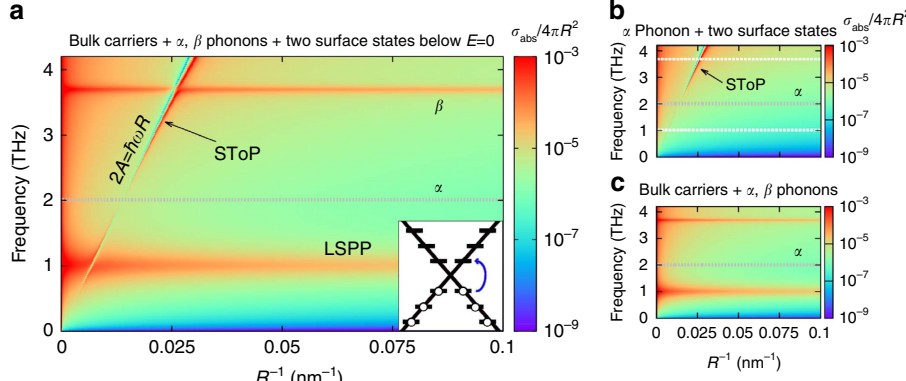

**Figure 3 | Absorption modified by topological surface states.** (**a**) When the topologically protected surface states are occupied up to the Dirac point, the two uppermost electrons (at energy $-A/R$) couple to empty states above the Dirac point (at $A/R$). As a result, the absorption cross-section contains an additional maximum (SToP, marked with an arrow) and a zero, which depend on the radius of the nanoparticle. The screening effect (along the line $2A = \hbar wR$) occurs when the energy spacing of surface states is matched by the incident radiation and the surface conductivity becomes infinite screening the bulk. The SToP mode arises, because the surface electron acts as a mediator of the interaction between light and the $\alpha$-phonon (grey dotted line), which otherwise does not absorb. This mode interacts with the localised surface plasmon (LSPP) and bulk $\beta$-phonon (the two horizontal modes) giving rise to the splittings. (**b**) In this figure the free carriers and $\beta$-phonon contributions to the bulk dielectric function were artificially removed, tht is, $w_{p\beta} = w_{pf} = 0$ (the absent modes marked by white dashed lines). This shows that the SToP mode (marked with an arrow) originates from the interaction of the $\alpha$-phonon with the surface states. (**c**) Cross-section of the nanoparticle without the effect of surface states.

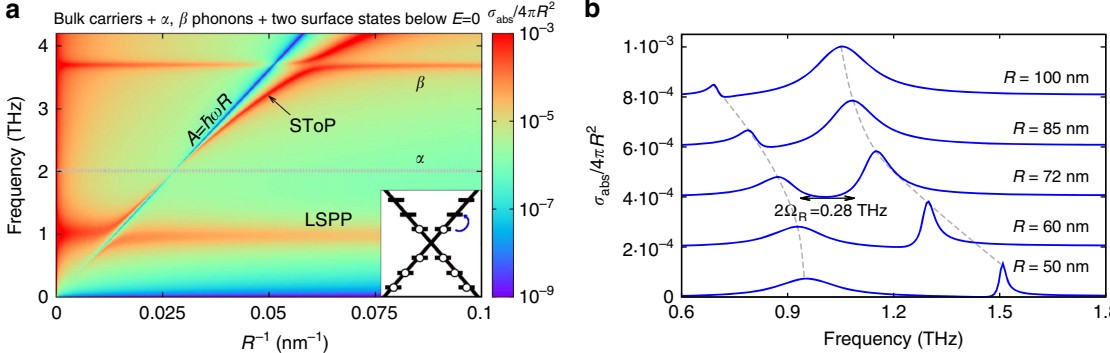

**Figure 4 | Effects of varying Fermi energy.** (**a**) Absorption cross-section of the TINP with two states immediately above the Dirac point occupied. The two uppermost electrons (at energy $A/R$) are perturbed by incident light, which causes them to couple to empty states higher in energy (at $2A/R$), leading to a line of suppressed absorption along $A = \hbar w R$ and a SToP mode (marked with an arrow). (**b**) For this Fermi energy, there is a significant Rabi splitting due to the interaction of the SToP mode and the localised surface plasmon (LSPP). When the two modes avoid each other most ($R = 72$ nm), the splitting $\Omega_R = 0.14$ THz is a considerable fraction of the average peak frequency ($> 0.1$), which is a signature of strong coupling.

view of possible applications for spintronics. As noted in ref. 10, the states close to the Dirac point can only scatter into other surface states due to the small energy range of phonons in the material[29]. Early theoretical studies found that the surface states couple to a single surface phonon developed from bulk $\alpha$ (2.0 THz)[30,31]. The coupling to a single optical phonon at $\sim 8$ meV (1.9 THz) was later confirmed with time-resolved angle-resolved photoemission spectroscopy[32] and transport measurements[33]. For a moment let us assume that only coupling to the optical phonon and other (for example, acoustic) phonons at lower energy occurs. In the worst case, this coupling will destroy any coherent effect of the surface states in bulk samples at finite temperatures. However, in the nanoparticles the states are discrete and the scattering can only occur if the phonon exactly matches the energy difference between the states. This implies that the effect of the surface states will be removed only in particles with radii below $R_T = 37.5$ nm (surface level spacing $A/R_T = 8$ meV) at finite temperatures (Supplementary Fig. 2). Even if other phonons contribute, the discrete nature of the surface states and the small energy range of phonons will strongly restrict the scattering. Moreover, the number of surface states decreases with radius further reducing the possibility of scattering in small nanoparticles (Supplementary Note 3). Finally, the states far from the Dirac point have bulk-like character and participate less due to the weaker overlap of wavefunctions[19]. Thus, it may be that the effects of topologically protected surface states on the nanoparticle can be observed at room temperature.

## Discussion

Using a simple model we have shown that the topologically protected surface states will have a strong effect on the optical properties of nanoparticles made from TIs compared with conventional insulators. Focusing on the nanoparticle made from $Bi_2Se_3$, we found two distinctive features. First, the electrons in surface states screen the bulk of the nanoparticle acting as a conducting layer at certain frequencies, which suggests their use as coatings. Second, an electron in the surface state mediates the interaction of the $\alpha$-phonon with light giving rise to the SToP mode. The energy of the mode can be tuned by varying the shape and size of the nanoparticle. This contrasts with the behaviour of other known modes in the far-infrared where the energy can only be changed by changing the material[34,35]. The mode leads to enhanced absorption and couples strongly to the LSPP.

The normalized coupling strength is comparable to the largest values achieved in the THz region in semiconductor quantum wells[36,37]. This, in addition to the large energy density stored in the SToP mode, will be of interest in cavity quantum electrodynamics (QED). Fast coherent exchange of energy characteristic of the strong coupling regime may be of use for THz lasers[38] and the screening effect—to make low-loss waveguides[39]. These in turn can be applied for sensing in the THz range[38,39]. In the limit of small number of photons it may also be important to give up classical description of light as was done here and instead treat it quantum mechanically as discussed in ref. 28. Finally, although pairs of degenerate states react in the same way to linearly and circularly polarized light, the individual states do differentiate between the two. This, together with the ability to manipulate the properties of the nanoparticle by varying the Fermi energy, will be of interest in the area of quantum information. Equally important is the fact that topological protection of the surface states suggests that the predicted effects will be resistant to surface passivation and may possibly be observed at room temperature. Experimentally, TINPs can fabricated with electron beam lithography[23] or using vapour–liquid–solid growth on gold nanoparticles[40]. A possible experimental setup to probe the described phenomena would be to study the absorption of the topological nanoparticles as a function of Fermi energy. In practice, the Fermi level can be varied by irradiating a TI with an electron beam[41].

To conclude, the present work provides a glimpse into the interaction of TI nanostructures with light where topological effects have been shown to manifest in a previously unknown way. The rich behaviour found contains a large potential to be realized and calls for more investigation.

**Data availability**. The data that support the findings presented and that have been used to produce the figures of this study are available from research data management system of Imperial College London and can be accessed with the following link: https://imperialcollegelondon.app.box.com/s/7cjxr035r97fc31fenwi1d9w1ggdzetx.

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

## Acknowledgements

G.S. was supported through a studentship in the Centre for Doctoral Training on Theory and Simulation of Materials at Imperial College London funded by EPSRC Grant Number EP/L015579/1. V.G. acknowledges support of Leverhulme Trust.

## Author contributions

V.G. proposed the initial idea and together with P.D.H. conceived the project. G.S. developed the analytical theory and performed calculations. G.S. and D.K.K.L. devised selection rules and finite temperature simulations. All authors analysed the results and wrote the manuscript. V.G. supervised the project.

## Additional information

**Competing financial interests:** The authors declare no competing financial interests.

