## [Peer review file · Nature Communications]

Reviewers' Comments:

Reviewer #1 (Remarks to the Author)

This manuscript presents some new numerical results of the THz wave absorption properties of a spherical nanoparticle made of Bi₂Se₃ topological insulator.

The key numerical result is:

Absorption cross sections showing strong hybridisations among LSP modes and phonon modes of bulk material and the surface modes (SToP), which are transitions among quantized (topological) surface states.

I think the method is acceptable for low intensity incident wave and small particle sizes. The methodology is summarised below. The authors used a set of fitting parameters (from the experimental in Ref. 19) for the dielectric function of the bulk Bi₂Se₃ (see Eq. 2). The authors have also used the parameter A/R in Ref. 2 of the supplementary materials (which was obtained from a $k \cdot p$ perturbation theory) to evaluate the eigenstates confined on a spherical surface (Eqs. 5 to 10 in Supplementary Materials. Then, within the linear approximation, the time-dependent surface charge density due to an oscillating driving electric-field is obtained by the transition rates (Eq. 14). The surface charge density is further used to evaluate the total dipole moment of the whole particle including the surface and bulk contributions in quasi-static approximation. The dipole moment is then used to find the absorption cross section of the particle.

The results could be useful for comparisons with experiments in the future and therefore should be published in some good journals. However, the presentation in this manuscript should be improved to better illustrate the physics behind. For example, the meaning of "Quantum Plasmons" in the title is not defined. It looks like that the "Quantum" part is about the discrete electronic states due to finite surface. It will be nice if the author can use better term to emphasize on the difference from quantum dots and quantization of plasmons.

Reviewer #2 (Remarks to the Author)

The manuscript under review predicts a novel light absorption resonance in topological nanoparticles. In a nutshell, it is argued that a nanoparticle made of an ordinary insulator will only display two (light) absorption resonances while one made of topological insulator will display an intermediate one due to the topological surface states.

I believe this theoretical effect is predicted in an oversimplified context and has very little chances to be observed experimentally. For this reason I cannot recommend the article for publication.

1. The theoretical model starts by placing a (single) Dirac metal on a sphere. In reality, the nanoparticle will be etched out of crystal, hence one will have to deal with the surface states on different surface terminations. It is known that Dirac cones are sensitive to surface direction and surface termination, hence the picture of a single Dirac metal on the surface of the sphere is, the least, an oversimplification.

2. The effect is relevant for nanoparticles of radius less than 100nm. The authors fail to say how such a nanoparticle will be fabricated out of Bi₂Se₃ which has a very large unit cell.

3. Recall that the metallic character of the surface states are protected by the bulk; if there is not enough "bulk," there cannot be any protection, hence the main assumption of topological protection is under question.

4. At a more in-depth level, the metallic character is generally guaranteed on flat surfaces where the back-scattering is prohibited, but on curved surfaces this is no longer the case. For this reason, I believe the Dirac metal will rather be a trivial Anderson insulator.

5. Given all the above, I believe only a first principle calculation will be convincingly enough on proving or disproving the effect. Perhaps, in a first phase one could investigate discrete lattice models terminated in various ways to give a nanoparticle.

I have other minor comments if the authors want to consider in the future. I think it is one of our main duties, as a community, to give proper acknowledgement to the works who initiated the fields mentioned in the introduction, such as [Phys. Rev. Lett. 100, 013904 (2008)] for initiating the field of topological photonic crystals and [Phys. Rev. Lett. 103, 248101 (2009)] for initiating the field of topological acoustic crystals.

Reviewer #3 (Remarks to the Author)

In this work the authors present a model study on the interaction of topological insulator nanostructures with the electromagnetic field. The results are novel and interesting and the paper is well written. The topic is timely and can potentially attract the interest of theoretical and experimental physicists in the fields of plasmonics and quantum condensed matter physics.

Before I can recommend publication on Nature Communications, the authors need to address the comments below:

i) According to the title this is a study on quantum plasmons. However quantum plasmonics is generally regarded as the field of research that involves the study of the quantum properties of light and its interaction with matter at the nanoscale (see e.g. Nature Physics 9, 329-340 (2013)). All the results presented in this paper have been derived within a semiclassical model where the TI is treated within a quantum model and the electromagnetic field is treated classically. Hence the title and also some description in the abstract and the text can be misleading and/or inappropriate.

ii) In the abstract, and in the text in different points the authors point out that the proposed phenomenon may pave the way for quantum behaviour to be observed at room temperature. The authors should justify better this statement. For example also exciton absorption lines in organic thin films or in molecular aggregates can be considered as quantum phenomena which can be observed at room temperature.

Most important, all the calculations presented by the authors have been performed at zero temperature. The room temperature behaviour could be very different. The authors should make the effort to present some result as a function of temperature.

I think that it is not sufficient to claim that the surface states have shown very little temperature dependence.

This is a key point, since the interest of experimentalists and the possibility for interesting applications strongly depend on the behaviours at nonzero and room temperatures.

Response to the reviewers

We are grateful to the reviewers for reading our manuscript carefully and providing such meaningful reviews. Their comments and criticisms have been helpful in revising the manuscript and improving its presentation.

Reviewer 1

The reviewer gives very positive evaluation of our work. Their only suggestion is to choose a more descriptive title which we have done in the revised manuscript.

The results could be useful for comparisons with experiments in the future and therefore should be published in some good journals. However, the presentation in this manuscript should be improved to better illustrate the physics behind. For example, the meaning of "Quantum Plasmons" in the title is not defined. It looks like that the "Quantum" part is about the discrete electronic states due to finite surface. It will be nice if the author can use better term to emphasize on the difference from quantum dots and quantization of plasmons.

We are very grateful to the reviewer for reading the manuscript and thank them for positive feedback. We fully agree with their comment and have changed the title to "Topological nanoparticle: single-electron induced surface plasmons" We have better explained the motivation behind it stressing the similarities with plasmons in metallic nanoparticles. A sentence contrasting the behaviour with that of non-topological quantum dots has been added.

Reviewer 2

The reviewer states that the predicted phenomenon is novel however they doubt that our model is rigorous and can be realised in practice proposing further calculations. Following up on the suggestion, we have performed the suggested calculations which show good agreement with our analytical model. We have fully addressed the reviewer's concerns adding numerous supporting evidence and mentioning available experimental techniques to produce topological nanoparticles.

I believe this theoretical effect is predicted in an oversimplified context and has very little chances to be observed experimentally. For this reason I cannot recommend the article for publication.

We understand the reviewer's point of view and have expanded the manuscript including the arguments for why our model is robust in prediction of phenomenon and experimentally accessible relying on the strong basis which exists in the literature. This is detailed below where we carefully address the reviewer's comments.

The theoretical model starts by placing a (single) Dirac metal on a sphere. In reality, the nanoparticle will be etched out of crystal, hence one will have to deal with the surface states on different surface terminations. It is known that Dirac cones are sensitive to surface direction and surface termination, hence the picture of a single Dirac metal on the surface of the sphere is, the least, an oversimplification...At a more in-depth level, the metallic character is generally guaranteed on flat surfaces where the back-scattering is prohibited, but on curved surfaces this is no longer the case. For this reason, I believe the Dirac metal will rather be a trivial Anderson insulator.

We agree that deviation from ideal spherical shape will modify the surface spectrum of the nanoparticle however it cannot remove the effect. The existence of the surface states is guaranteed by the non-trivial Z_2 invariant of the material. On a flat surface they form a Dirac cone. As noted by Haldane [1] (and references therein), the Dirac cone covers area $A_{\mathbf{k}}$ which is a small fraction of the 2D 1BZ. This implies that in real space their wavefunctions cannot be localised over areas smaller than $\approx (2\pi)^2/A_{\mathbf{k}}$ rendering them insensitive to atomic size details of the surface. This is even more so for the nanoparticle surface where few discrete states available cannot produce fine features in real space. On a general surface, the surface states are influenced by surface curvature and surface termination. The curvature enters the employed model explicitly through Berry phase [2]. Also, the Dirac cone is only moderately distorted on different surfaces as has been shown in the work of Zhang, Kane and Mele [3] whose results we present in Fig. 1 below for convenience. More generally, the new resonance

is not restricted to spheres in the same way as ordinary plasmon resonance occurs in spherical metallic particles but also particles of other shapes [4]. The advantage of spherical geometry is that it allows analytic solution and provides physical insight, the effect however will remain for nanoparticles of other shapes. The information above has also been added in the revised manuscript.

Figure 1: Left: Dirac cone distortion on different crystal faces; angle θ is measured w.r.t. c -axis. Right: vertical shift of Dirac cone on (001) surface under the influence of various surface impurities. Figs. 2 and 3 in [3].

The effect is relevant for nanoparticles of radius less than 100nm. The authors fail to say how such a nanoparticle will be fabricated out of Bi_2Se_3 which has a very large unit cell... Recall that the metallic character of the surface states are protected by the bulk; if there is not enough "bulk," there cannot be any protection, hence the main assumption of topological protection is under question.

We completely agree with the reviewer that producing spherical nanoparticles is challenging but as described above the reported phenomenon relies on the nanoscale size of the particle rather than its shape. Though Bi_2Se_3 has a large unit cell, the resolution lies in its hexagonal layered structure ($c=2.864$ nm and $a=0.414$ nm [5]). Along the c -axis it consists of three identical quintuple layers (≈ 1 nm) shifted relative to one another (ABC-stacking). Thus a sphere with 10 nm radius is 20 quintuple layers high and up to 50 lattice parameters wide which should provide enough 'bulk' for protection. Looking from another point of view, the protection occurs for particle sizes much larger than the decay length of surface states into the bulk. Decay length of 0.5 nm for the model employed (Eq. 15 in [2]) agrees well with experimental evidence that states on opposite surfaces of thin films noticeably interact only for thicknesses below 5 nm [6]. The above suggests that the radius of 10 nm used for all absorption calculations is a reasonable limit (this information has been added in the revised manuscript). In practice, such nanoparticles can be produced with two techniques:

- Vapour-liquid-solid growth technique on gold nanoparticles yields Bi_2Se_3 nanowires of thicknesses down to 25 nm [7]. Stopping the growth early would enable one to obtain nanoparticles.
- Electron-beam lithography technique allows resolution of < 10 nm [8]. It is being used extensively in nanophotonics and has already been applied to fabricate Bi_2Se_3 microribbons. [9].

Given all the above, I believe only a first principle calculation will be convincingly enough on proving or disproving the effect. Perhaps, in a first phase one could investigate discrete lattice models terminated in various ways to give a nanoparticle.

We certainly agree that first principles calculations would strengthen our case. Such calculations exist for the case of topological insulator cylinder [10] and are shown in Fig. 2 (Left) for convenience. In addition, we have applied a tight-binding model [2] to the spherical nanoparticle. The results presented in Fig. 2 (Right) show that analytical model adequately describes states close to the Dirac point especially for large radii. Furthermore, the dispersion is weakly modified if one instead considers a cubic particle as expected from the topological nature of the states. These results have been added to the revised manuscript with details described in supporting information. Unfortunately, even the nanoparticles of smallest sizes considered would consist of thousands of atoms and are too costly to investigate with density functional theory. We certainly agree that our model has limitations but this is the best that can be done currently. Further work employing ambitious density functional theory calculations using dedicated large-scale simulation methods (such as ONETEP) would indeed

be interesting, but ultimately the validation can only from experiment, so there is value in making this result available to the community at this stage.

Figure 2: Left: Energy of surface state closest to the Dirac point for Bi_2Se_3 nanowire using three different approaches: tight-binding model (red squares), numerical diagonalisation of low-energy Hamiltonian (black circles) and analytical solution (blue dashed curve). Fig. 3 in [10]. Center: Same for a nanoparticle: tight-binding results for cubic (green squares) and spherical (blue circles) shapes together with the analytic solution (red line). Dotted horizontal line shows the bulk gap; the model breaks down in grey shaded area. Cube size is $2\sqrt[3]{4\pi R^3/3}$. The models are illustrated on the side.

I have other minor comments if the authors want to consider in the future. I think it is one of our main duties, as a community, to give proper acknowledgement to the works who initiated the fields mentioned in the introduction, such as [Phys. Rev. Lett. 100, 013904 (2008)] for initiating the field of topological photonic crystals and [Phys. Rev. Lett. 103, 248101 (2009)] for initiating the field of topological acoustic crystals.

We agree and have added references to the papers mentioned.

Reviewer 3

The reviewer rated the results and their presentation positively. They suggest publishing the manuscript on condition of improving the title and adding finite temperature simulations. Following up on the suggestion we have performed the simulations and have fully addressed all of the reviewer's concerns.

The results are novel and interesting and the paper is well written. The topic is timely and can potentially attract the interest of theoretical and experimental physicists in the fields of plasmonics and quantum condensed matter physics.

We thank the reviewer for high evaluation of the results and are very grateful for their comments which have been fully addressed as detailed below.

i) According to the title this is a study on quantum plasmons. However quantum plasmonics is generally regarded as the field of research that involves the study of the quantum properties of light and its interaction with matter at the nanoscale (see e.g. Nature Physics 9, 329-340 (2013))... Hence the title and also some description in the abstract and the text can be misleading and/or inappropriate.

We agree and have changed the title to "Topological nanoparticle: single-electron induced surface plasmons" We have also modified abstract and relevant text stressing the similarities with plasmons in metallic nanoparticles. The classical treatment of light is made more explicit and we have added a reference to the mentioned paper.

ii) In the abstract, and in the text in different points the authors point out that the proposed phenomenon may pave the way for quantum behaviour to be observed at room temperature. The authors should justify better this statement... Most important, all the calculations presented by the authors have been performed at zero temperature. The room temperature behaviour could be very different. The authors should make the effort to present some result as a function of temperature.

We agree and have expanded the relevant part of the manuscript adding a figure with finite-temperature simulations in the supporting information. These modifications are outlined below.

Figure 3: Left: (a) and (b): phonon energies determine possible states for scattering (c): probability of scattering for the helical states (bottom): Dirac cone as a function of T. Fig. 1 in [11]. Right: (a) and (b): Fermi surface and Dirac cone at 18 K (c): Dirac cone at 255 K (d): Photoemission intensity as a function of T at Fermi level (e): Same along FK direction (f): Same at Γ point. Fig. 1 in [12].

Presence of Dirac surface states at room temperature is confirmed in ARPES studies as presented in Fig. 3 for convenience. States close to the Dirac point can only scatter into other surface states due to small energy range of phonons in Bi_2Se_3 [13] as shown in Fig. 3(a). Early studies found that the surface states couple to the surface phonon developed from bulk α (2.0 THz) [14, 15]. The coupling to a single optical phonon at ≈ 8 meV (1.9 THz) was confirmed with:

- Time-resolved ARPES on (001) surface at 40 K [16]
- Transport measurement on elongated crystals at 30 K [17]

Scattering cannot happen if the energy spacing of the surface states is larger than the energy of available phonons. To model this we assume that the surface states with energy spacing ≥ 8 meV are unaffected by finite temperature while those with less are completely smeared out. This is a very strict approximation which also acknowledges the possibility of scattering by phonons of lower energy (e.g. acoustic). Finite temperature enters our model by multiplying δ_R (the surface term) with a smooth step function centered at $R_T = 37.5$ nm (surface level spacing $A/R_T = 8$ meV). We use the bulk dielectric function, ϵ_{in} measured at 300 K from a different source [9]. The resulting absorption cross section presented in Fig. 4 is valid for $T \leq 40$ K. This is a conservative estimate because small number of the surface states available in nanoparticle compared to bulk samples reduces scattering. In addition, the states far from the Dirac point have bulk-like character and participate less in surface scattering [1]. Thus the possibility of observing the effect at room temperature cannot be ruled out, however, without direct experimental evidence we have relaxed our claims in the revised manuscript.

Figure 4: Absorption cross-section of a TINP at finite temperature. For radii $R^{-1} < R_T^{-1} = (37.5 \text{ nm})^{-1}$ (vertical dashed line) the effect of surface states is smeared by electron-phonon coupling.

References

- [1] Titus Neupert, Stephan Rachel, Ronny Thomale, and Martin Greiter. Interacting Surface States of Three-Dimensional Topological Insulators. *Physical Review Letters*, 115(1):017001, June 2015.
- [2] Ken-Ichiro Imura, Yukinori Yoshimura, Yositake Takane, and Takahiro Fukui. Spherical topological insulator. *Phys. Rev. B*, 86(23):235119, December 2012.
- [3] Fan Zhang, C. L. Kane, and E. J. Mele. Surface states of topological insulators. *Phys. Rev. B*, 86(8):081303, August 2012.
- [4] Vincenzo Giannini, Antonio I. Fernandez-Domnguez, Susannah C. Heck, and Stefan A. Maier. Plasmonic Nanoantennas: Fundamentals and Their Use in Controlling the Radiative Properties of Nanoemitters. *Chemical Reviews*, 111(6):3888–3912, June 2011.
- [5] Bismuth (Bi₂Se₃) selenide crystal structure, chemical bond, lattice parameter (including data of related compounds). (41C):1–4, 1998. DOI: 10.1007/10681727_955.
- [6] Yi Zhang, Ke He, Cui-Zu Chang, Can-Li Song, Li-Li Wang, Xi Chen, Jin-Feng Jia, Zhong Fang, Xi Dai, Wen-Yu Shan, Shun-Qing Shen, Qian Niu, Xiao-Liang Qi, Shou-Cheng Zhang, Xu-Cun Ma, and Qi-Kun Xue. Crossover of the three-dimensional topological insulator Bi₂Se₃ to the two-dimensional limit. *Nat Phys*, 6(8):584–588, August 2010.
- [7] Hailin Peng, Keji Lai, Desheng Kong, Stefan Meister, Yulin Chen, Xiao-Liang Qi, Shou-Cheng Zhang, Zhi-Xun Shen, and Yi Cui. Aharonov-Bohm interference in topological insulator nanoribbons. *Nature Materials*, 9(3):225–229, March 2010.
- [8] Yifang Chen. Nanofabrication by electron beam lithography and its applications: A review. *Microelectronic Engineering*, 135:57–72, March 2015.
- [9] P. Di Pietro, M. Ortolani, O. Limaj, A. Di Gaspare, V. Giliberti, F. Giorgianni, M. Brahlek, N. Bansal, N. Koirala, S. Oh, P. Calvani, and S. Lupi. Observation of Dirac plasmons in a topological insulator. *Nature Nanotechnology*, 8(8):556–560, August 2013.
- [10] R. Egger, A. Zazunov, and A. Levy Yeyati. Helical Luttinger Liquid in Topological Insulator Nanowires. *Physical Review Letters*, 105(13):136403, September 2010.
- [11] Z.-H. Pan, A. V. Fedorov, D. Gardner, Y. S. Lee, S. Chu, and T. Valla. Measurement of an Exceptionally Weak Electron-Phonon Coupling on the Surface of the Topological Insulator Bi₂Se₃ Using Angle-Resolved Photoemission Spectroscopy. *Physical Review Letters*, 108(18):187001, May 2012.
- [12] Richard C. Hatch, Marco Bianchi, Dandan Guan, Shining Bao, Jianli Mi, Bo Brummerstedt Iversen, Louis Nilsson, Liv Hornekr, and Philip Hofmann. Stability of the Bi₂Se₃(111) topological state: Electron-phonon and electron-defect scattering. *Physical Review B*, 83(24):241303, June 2011.
- [13] W. Richter and C. R. Becker. A Raman and far-infrared investigation of phonons in the rhombohedral V₂VI₃ compounds Bi₂te₃, Bi₂se₃, Sb₂te₃. *Phys. Stat. Sol. (b)*, 84(2):619–628, December 1977.
- [14] Xuetao Zhu, L. Santos, R. Sankar, S. Chikara, C. . Howard, F. C. Chou, C. Chamon, and M. El-Batanouny. Interaction of Phonons and Dirac Fermions on the Surface of Bi₂Se₃: A Strong Kohn Anomaly. *Phys. Rev. Lett.*, 107(18):186102, October 2011.
- [15] Xuetao Zhu, L. Santos, C. Howard, R. Sankar, F. C. Chou, C. Chamon, and M. El-Batanouny. Electron-Phonon Coupling on the Surface of the Topological Insulator Bi₂Se₃ Determined from Surface-Phonon Dispersion Measurements. *Physical Review Letters*, 108(18):185501, May 2012.

- [16] J. A. Sobota, S.-L. Yang, D. Leuenberger, A. F. Kemper, J. G. Analytis, I. R. Fisher, P. S. Kirchmann, T. P. Devereaux, and Z.-X. Shen. Distinguishing bulk and surface electron-phonon coupling in the topological insulator Bi_2Se_3 using time-resolved photoemission spectroscopy. *Phys. Rev. Lett.*, 113:157401, Oct 2014.
- [17] M. V. Costache, I. Neumann, J. F. Sierra, V. Marinova, M. M. Gospodinov, S. Roche, and S. O. Valenzuela. Fingerprints of inelastic transport at the surface of the topological insulator Bi_2Se_3 : Role of electron-phonon coupling. *Phys. Rev. Lett.*, 112:086601, Feb 2014.

Reviewers' Comments:

Reviewer #2 (Remarks to the Author)

Dear Editor

the authors have address all my comments in a satisfactory way. I believe the paper is now sound and will present interest in our community. I recommend the paper for publication.

Reviewer #3 (Remarks to the Author)

The authors have largely improved the manuscript.

Specifically, they changed the title and the text avoiding misleading sentences; they have qualitatively described the expected higher temperature behavior; they included a tight binding calculation of the spectrum, enforcing their model in the supplemental file.

Now, in view of the novelty and interest of the proposed phenomenon and of the manuscript changes, I can recommend publication on Nature Communications